# Exploring Cold Hardiness within a Butterfly Clade: Supercooling Ability and Polyol Profiles in European Satyrinae

**DOI:** 10.3390/insects13040369

**Published:** 2022-04-09

**Authors:** Pavel Vrba, Alena Sucháčková Bartoňová, Miloš Andres, Oldřich Nedvěd, Petr Šimek, Martin Konvička

**Affiliations:** 1Biology Centre of Czech Academy of Sciences, Institute of Entomology, Branišovská 31, 370 05 České Budějovice, Czech Republic; vrba_pavel@centrum.cz (P.V.); al.bartonova@gmail.com (A.S.B.); nedved@prf.jcu.cz (O.N.); simek@bclab.eu (P.Š.); 2Faculty of Science, University of South Bohemia, Branišovská 31, 370 05 České Budějovice, Czech Republic; 3JARO Jaroměř, Národní 83, 551 01 Jaroměř, Czech Republic; fagus1@seznam.cz

**Keywords:** butterfly physiology, carbohydrate, cryoprotectants, elevation, cold hardiness, Lepidoptera: Nymphalidae, mountains, temperate zone, winter survival

## Abstract

**Simple Summary:**

In insects distributed in temperate and cold zones, cold hardiness during overwintering crucially affects the distribution, including range shifts due to climate change. Our previous work on the genus *Erebia*, a cold-adapted and species-rich group of the sub-family Satyrinae (Nymphalidae), disclosed unexpected diversity of cold hardiness strategies, with closely related species surviving or not surviving freezing of larval body fluids. Asking whether this diversity is peculiar to this genus, or may be common in the Satyrinae clade, we investigated supercooling ability, contents of sugars and polyols in overwintering larvae tissues, and evolutionary signal of these traits of eight European Satyrinae species (from seven genera) and compared them with the *Erebia* representatives investigated earlier. We show that cold hardiness strategies are indeed diverse in the group and that high mountain and continental steppe species employ similar cryoprotection mechanisms, differing from those employed by species of more mesic environments.

**Abstract:**

The cold hardiness of overwintering stages affects the distribution of temperate and cold-zone insects. Studies on *Erebia*, a species-rich cold-zone butterfly genus, detected unexpected diversity of cold hardiness traits. We expanded our investigation to eight Satyrinae species of seven genera. We assessed Autumn and Winter supercooling points (SCPs) and concentrations of putatively cryoprotective sugars and polyols via gas chromatography–mass spectrometry. *Aphantopus hyperantus* and *Hipparchia semele* survived freezing of body fluids; *Coenonympha arcania*, *C. gardetta,* and *Melanargia galathea* died prior to freezing; *Maniola jurtina*, *Chazara briseis,* and *Minois dryas* displayed a mixed response. SCP varied from −22 to −9 °C among species. Total sugar and polyol concentrations (TSPC) varied sixfold (2 to 12 μg × mg^−1^) and eightfold including the *Erebia* spp. results. SCP and TSPC did not correlate. Alpine *Erebia* spp. contained high trehalose, threitol, and erythritol; *C. briseis* and *C. gardetta* contained high ribitol and trehalose; lowland species contained high saccharose, maltose, fructose, and sorbitol. SCP, TSPC, and glycerol concentrations were affected by phylogeny. Species of mountains or steppes tend to be freeze-avoidant, overwinter as young larvae, and contain high concentrations of trehalose, while those of mesic environments tend to be freeze-tolerant, overwinter as later instars, and rely on compounds such as maltose, saccharose, and fructose.

## 1. Introduction

The recent debates on the effects of warming climate on biotic communities raise interest in the cold-adapted insects inhabiting narrow cold-climate zones, such as those in high mountains [1,2,3]. Due to warming climate, cold-adapted species may be impaired by habitat changes, such as ascending timberline in species dependent on open grassland conditions [4], or by direct climatic effects. These may operate during the adult season, such as heat waves affecting adult survival [5], but also in the cold season, such as decreasing snow amount resulting in worse insulation of overwintering stages [6,7], or temporary snowmelts causing, e.g., diapause disruptions and subsequent mortality [8].

As there are a few studies exploring cold hardiness strategies on a larger spectrum of closely related species differing in climatic and habitat preferences, we targeted, in our previous studies, overwintering larvae of the butterflies of the genus *Erebia* Dalman, 1816 (Nymphalidae: Satyrinae). This large Holarctic genus (≈100 species described to date) is tightly associated with mountain environments, and a good background knowledge on its life history, population structure, and phylogeny exists [9,10,11,12,13]. Results obtained on four species indicated a counterintuitive clinal pattern, in which a lowland species survived deep (−21 °C) freezing of body fluids, while three alpine species did not survive the freezing and died at considerably higher temperatures [14]. We initially interpreted this by insulation properties of the snow layer, more reliable in alpine elevations than in temperate zone lowlands. Further work with a higher number of species [15] complicated the matter. Values of the supercooling point (the temperature of body fluids freezing: SCP) did not correlate with concentrations of putatively cryoprotective soluble carbohydrates, i.e., sugars and polyols; both SCP values and the concentrations varied with acclimation conditions experienced by the late autumn larvae, and the chemical identity of the putatively cryoprotective compounds varied greatly among the congeneric species. In one species, part of the larvae survived and another part died at temperatures below SCP, indicating a mixed situation seldom observed in Lepidoptera [16].

This diversity of traits associated with overwintering existing in *Erebia* spp. could be peculiar to this cold-adapted genus, which rapidly radiated in climatically variable and often harsh conditions of Neogene/Quaternary Palearctic grasslands [12,17], and display remarkable diversity, e.g., in preferred microclimates [18]. Alternatively, such diversity can exist in temperate and cold-zone butterflies in general. To explore the latter possibility, we extended our investigation towards a wider selection of species from the sub-family Satyrinae, a group developing on high-silica-containing monocotyledons, including some 2400 species world-wide, and inhabiting a wide range of temperate zone habitats. 

Regarding general overwintering abilities [19,20,21,22], insects exposed to winter subzero temperatures may survive crystallization of body liquids (“freeze-tolerant species”, their lower lethal temperature (LLt) < SCP). “Freeze-avoidant species”, in which LLt ≥ SCP, cannot survive freezing of body liquids. Finally, “chill-susceptible species” die at temperatures well above the SCP, depending on the duration of exposure to low temperatures, their physiological state, and other factors [23]. Whereas species-specific SCP values can be determined with a rather small amount of material (≈10 individuals per species) [24], measuring LLt uses hundreds of individuals, which is not feasible with rare species originating from extreme habitats, or with species of conservation concern. Still, even the sole knowledge of the SCP provides basic information on winter survival strategies. Further, surviving cold temperatures may be facilitated by the low-molecular sugars and polyols, which decrease the SCP and contribute to the protection of cells and proteins [25,26]. Knowledge of their specific roles remains fragmentary; some substances are accumulated in high concentrations during winter and colligatively decrease the SCP [19], whereas others occur at low concentrations and protect membranes and macromolecules [26,27]. In Lepidoptera, metabolism is reprogrammed to the production of low-molecular cryoprotectants, primarily central metabolites such as glucose or trehalose, and further extended to polyhydroxylated alcohols involving triose, tetrose, and pentose structure, typically glycerol and sorbitol [15]. 

We investigated eight non-*Erebia* Satyrinae overwintering butterfly larvae, all of them univoltine and occurring in Central Europe, in which we determined SCP and measured sugar and polyol contents. Throughout the paper, we compare the findings with the earlier *Erebia* spp. results [14,15]. We asked the following questions:What are the supercooling abilities and concentrations of putatively cryoprotectant compounds in the studied species, do these measures of cold hardiness correlate with each other, and where do the studied species stand relative to the high-elevation *Erebia* spp.?How do the above characteristics of cold hardiness change from late autumn to winter?Which of the sugars and polyols, based on correlations between their concentration and cold hardiness measures, play a cryoprotective role in the studied species?Is there a relationship among the Satyrinae cold hardiness characteristics and their vertical distribution (the latter standing for the harshness of conditions experienced by overwintering larvae)?Is there a phylogenetic signal of the identity of cryoprotectant compounds used?

## 2. Materials and Methods

### 2.1. Study Species

The eight species we investigated (Table 1) all develop on grasses and typically form a single generation per annum. 

*Aphantopus hyperantus* (Linnaeus, 1758): A Palearctic species, distributed from the Pyrenean peninsula to temperate China and Korea. In Europe, it reaches South Scandinavia and vertically occurs from the lowlands to the mountain zone (Czech Republic: <1200 m a.s.l.). Its larvae retain feeding activity during mild winters. 

*Chazara briseis* (Linnaeus, 1764): Ponto-Mediterranean species, distributed from NW Africa to Central Asia, reaching Eastern Germany to the North. North of the Alps, its habitat is open-turf dry grasslands. The current distribution is severely fragmented [28,29]. In captive rearing, we observed larval feeding even during mild winters. 

*Coenonympha arcania* (Linnaeus, 1761): European species, distributed from Western Europe to Asia Minor, the Caucasus and Ural Mts. Inhabitant of open woodlands, forest mantles and edges, requiring mosaics of grasslands and shrubs. A partial second generation appears in warm years [30]. 

*Coenonympha gardetta* (de Prunner, 1798): A sister taxon of the above [31]. It is a subalpine species restricted to the Alps, where it inhabits biotopes near the timberline. 

*Hipparchia semele* (Linnaeus, 1758): European species with a prominently oceanic distribution [32]. Its range follows coastal areas from the Baltic countries through the British Isles to the Mediterranean region and Southern Russia. Populations in Central Europe are declining [33].

*Maniola jurtina* (Linnaeus, 1758): A west-Palearctic species, distributed from Northwestern Africa and the British Isles to Western Siberia and Iran. A generalist occurring from the lowlands to the mountain zone. Its European distribution reaches South Scandinavia latitudinally and ca 1000 m a.s.l. vertically. Widespread and common.

*Melanargia galathea* (Linnaeus, 1758): A west-Palearctic species, distributed from NW Africa to the Urals, Asia Minor and Transcaucasia; it is absent in more northerly areas (e.g., Scandinavia). 

*Minois dryas* (Scopoli, 1763): A Euro-Siberian species, distributed from the Pyrenees across Europe (except southern peninsulas) and as far east as Korea and Japan. The distribution in Central Europe is discontinuous and includes both xeric and humid grasslands [34].

In addition, we refer here to earlier *Erebia* spp. results (Table 1). Vrba et al. [15] studied cold hardiness and putative cryoprotectants in *E. medusa* (Denis and Schiffermüller, 1775), a lowland freeze-tolerant species; *E. aethiops* (Esper, 1777), a submontane species (<1500 m a.s.l. in the Alps) with a mixed strategy; and three freeze-avoidant high-elevation species, *E. pronoe* (Esper, 1780) (subalpine grasslands), *E. cassioides* (Reiner and Hochenwarth, 1792) (sparsely vegetated rocky substrates), and *E. pluto* (De Prunner, 1798) (screes of alpine and subnivean zone). A still earlier study [14] targeted *E. medusa* (as above); *E. sudetica* (Staudinger, 1861), a subalpine species of timberline grasslands; *E. epiphron* (Knoch, 1783), an alpine grasslands species; and *E. tyndarus* (Esper, 1781), an alpine species of rocky substrates. 

### 2.2. Captive Rearing

The pre-hibernating larvae of *A. hyperantus*, *C. arcania*, *C. gardetta*, *M. jurtina*, and *M. galathea* were obtained from wild-caught females. Larvae of *C. briseis*, *H. semele*, and *M. dryas* originated from an ex situ conservation rearing ([29], Table 1). In both cases, the females oviposited in outdoor cages (wire frame 50 cm × 50 cm × 100 cm covered by nylon mesh) located in a half-shaded garden corner in České Budějovice, the Czech Republic (48°58′ N, 14°28′ E, 400 m a.s.l.) (the wild-caught species) and in a similar facility in Barchov, Czech Republic (50°12′ N, 15°34′ E, 250 m a.s.l.) (the captive-reared species). A maximum of five females were placed into one cage, the cages contained potted grass used as host plants at the localities. All the females readily accepted feeding by saccharose solution.

The larvae were taken from these rearing facilities immediately prior to experimental treatments, which were Autumn (November 2018) and Winter (January 2019). The Autumn treatment targeted the effects of the first frosts, the Winter treatment aimed at the highest cold hardiness level in the coldest month.

Identical rearing but different acclimation conditions were used in the earlier *Erebia* studies (cf. Table 1). Vrba, Konvicka, and Nedved [14] acclimated the larvae to a constant 5 °C, and Vrba et al. [15] used treatments simulating early autumn, late autumn, and winter. Here, we consider only late autumn and winter results (identical with Autumn and Winter here).

### 2.3. Supercooling Point and Polyol Profiles 

The larvae were weighed and used for supercooling point measurement. SCP was measured individually using a PICO recorder with hand-made type K thermocouples [35], attached to the body of the experimental caterpillars enclosed in a syringe [36]. Larvae were gradually cooled above liquid nitrogen and the cooling rate was controlled at 1 °C/min. After an exotherm appeared on the recorder, the larva was kept in the cooling device until its body temperature decreased again to the crystallization temperature and then removed and warmed up at room temperature. Observation of movement of warmed-up caterpillars enabled us to determine if they were freeze-tolerant. Appendix A contains the original data.

The polyols were measured by gas chromatography–mass spectrometry (GC–MS) on a DSQ mass spectrometer (Thermo Scientific, Waltham, MA, USA). Following [37], the larvae were weighed, stored frozen at −80 °C, thawed before the assay, homogenized in 70% ethanol, and the extract centrifuged. The supernatant was defatted by hexane, then dried and treated with O-methylhydroxylamine (80 °C for 30 min) and trimethylsilylimidazol (80 °C for 30 min). After re-extraction into 100 µL isooctane, a 1 µL aliquot was separated on a 30 m × 0.25 mm × 0.25 µm DB-1MS capillary column (Agilent, Santa Clara, CA, USA). The MS conditions: EI source 220 °C, 70 eV; helium flow-rate 1.1 mL/min; GC inlet 250 °C; splitless injection 1.3 min. The temperature program: 80 °C hold, 1 min, 20 °C/min to 180 °C, 5 °C/min to 200 °C, 25 °C/min to 300 °C hold for 3 min; transfer line 280 °C. The acquired data were processed by Thermo Xcalibur 2.1 software. The following metabolites were identified using the retention time, EI mass spectrum, and quantified by external calibration against the corresponding standard: glycerol, ribose, arabinitol, ribitol, fructose, glucose, mannitol, sorbitol, scyllo-inositol, myo-inositol, saccharose, trehalose, and maltose.

### 2.4. Statistical Analyses

Two-way ANOVA with species, treatment, and species x treatment interaction effects were used to compare SCPs, and the total concentrations of sugars and polyols (TSPCs) across the eight non-*Erebia* Satyrinae. An additional nine (SCPs) and five (TSPCs) *Erebia* species were assayed under somehow different acclimation conditions (Table 1); we compared only means and standard deviations of the earlier results.

Pearson’s correlations were used to investigate relations between SCP, the elevation of origin (Table 1), TSPCs, and concentrations of individual sugars and polyols. The correlations were computed for both treatments, for Autumn and Winter separately, and after adding the *Erebia* results. 

The composition of sugars and polyols were analyzed using multivariate statistics, the canonical correspondence analysis (CCA) in CANOCO [38]. CCA ordinates samples according to their composition and constrains the ordination according to predictors of interest. The numeric response variables were contents of individual compounds (log-transformed), whereas the factorial predictors were individual species. We ran the analyses with centering by species and samples and tested their significances using 999 Monte Carlo permutations. 

### 2.5. Phylogeny of Sugar and Polyol Profiles

We used a phylogenetic tree of the studied species (Appendix B) to detect a possible phylogenetic signal in the sugar and polyol concentrations and the supercooling point of the larvae. 

We computed, separately for Autumn and Winter, Blomberg’s K [39] and Pagel’s λ in R package “phytools” [40] for the major compounds (glycerol, fructose, glucose, sucrose, and trehalose), SCP and TSPC. These statistics compare the observed signal in a trait (a log-transformed in case of concentrations) to the signal under a Brownian motion model of trait evolution on a phylogeny. Blomberg’s K is based on mean square errors, while Pagel’s λ transforms phylogeny to fit the trait data. In both statistics, the values ≈ 0 correspond to a random or convergent evolution, while values ≈ 1 (both statistics) or >1 (Blomberg’s K) indicate a phylogeny dependency. If a phylogenetic signal was present, we reconstructed the ancestral states on individual nodes of the tree using Felsenstein’s phylogenetic independent contrasts (PICs) using the ace function in R “ape” package [41].

### 2.6. Sugar and Polyol Profiles versus Cold Hardiness

The fourth-corner analysis [42] relates three data tables, one with identity of the samples (here, butterfly species), one with the samples’ properties (i.e., sugar and polyol concentrations), and one with species traits. The species traits potentially related to overwintering obtained from this study were: cold hardiness strategy (three states: freeze-avoidant, freeze-tolerant, mixed), SCP, TCSP (both continuous numeric: means across both treatments used for simplicity), overwintering larval instar (three states: 1—freshly hatched, 2—medium instars, 3—grown-up larvae before pupation), and elevation of sampling localities (continuous numeric; Table 1). In CANOCO, this analysis proceeded in four steps: (1) CCA constraining sugar and polyol composition ~species|treatment; (2) the matrix of phylogenetic distances (from Appendix B) imported and subject to principal coordinate analysis (PcoA); (3) PcoA scores used to constrain CCA axes from the first step (control for phylogeny); (4) the resulting axes constrained by functional traits via redundancy analysis (RDA), a multivariate variant of linear regression. We used the forward-selection procedure to select the best fitting combination of traits. 

## 3. Results

### 3.1. Supercooling Ability and Sugar and Polyol Concentrations

In both Autumn and Winter treatments, all *A. hyperantus* and *H. semele* larvae survived freezing of body fluids (freeze-tolerant species). No *C. arcania*, *C. gardetta,* or *M. galathea* larva survived the freezing (freeze-avoidant species). Mixed situations applied for *M. jurtina*, in which 5/1 larvae (Autumn/Winter treatments, of 16/16 individuals), *C. briseis*, in which 0/2 larvae (of 15/10), and *M. dryas*, in which 2/0 larvae (of 15/8) survived the freezing. In the earlier *Erebia* studies, *E. medusa* was freeze-tolerant, *E. aethiops* displayed a mixed response (3/5 of 16/16 survived), and the six remaining species were freeze-avoidant [14,15].

Supercooling points (Figure 1A, Appendix A) were, in Autumn, below −20 °C for the freeze-avoidant species *C. arcania* and *C. gardetta*, as well as for the mixed-response *C. briseis* and *M. dryas*. They were near −10 °C for the other mixed-response *M. jurtina* and freeze-avoidant *M. galathea*. Freeze-tolerant *H. semele* and *A. hyperantus* displayed values above −10 °C. The values were similar in Winter, except for the two mixed-strategy species (*C. briseis* and *M. jurtina*), in which the SCP converged, i.e., increased in *C. briseis* and decreased in *M. jurtina*. Two-way ANOVA corroborated significant differences among species (F_(7,215)_ = 75.21, *p* < 0.0001), no difference between treatments (F_(1,215)_ = 0.06, *p* = 0.81), and significant species × treatment interaction (F_(7,215)_ = 3.75, *p* < 0.001). In the *Erebia* studies, SCP values were close to −20 °C in *E. cassioides* in Autumn and Winter and *E. pronoe* in Autumn. *Erebia pluto* and *E. aethiops* reached an SCP near −15 °C. 

For total sugar and polyol concentrations (Figure 1B) in Autumn, *M. galathea* and *M. dryas* reached the highest values, both >10 μg × mg^−1^. Lower values near 5 μg × mg^−1^ were found in *A. hyperantus*, *C. gardetta*, *C. briseis*, and *H. semele*, while *M. jurtina* and *C. arcania* displayed the lowest values. In Winter, no species increased TSPC, *M. galathea* and *M. dryas* dropped it to ≈5 μg × mg^−1^, and weak decreases occurred in *A. hyperantus*, *C. briseis*, and *C. gardetta* (two-way ANOVA, species: F_(7,142)_ = 45.17, *p* < 0.0001; treatments: F_(1,142)_ = 68.49, *p* < 0.0001; species x treatment interaction: F_(7,142)_ = 19.76, *p* < 0.0001). In the earlier *Erebia* studies, high concentrations (≈15 μg × mg^−1^) were found in *E. pluto* (Autumn), *E. cassioides* (both treatments), and *E. aethiops* and *E. pronoe* (Winter). In the latter two species, Autumn concentrations were much lower (<5 μg × mg^−1^). 

According to the general levels across the eight species assayed here (ANOVA, F_(12,195)_ = 11.0; *p* < 0.001 and LSD post hoc test in Appendix A), the sugars and polyols can be divided into three categories: major (glucose, saccharose), intermediate (fructose, trehalose), and minor. For the *Erebia* set [15] there was one major compound, trehalose, and three intermediate ones, glycerol, glucose, and saccharose. 

Mean SCPs and TSPCs (log-transformed for normality) did not correlate (Autumn: r = 0.14, t_(*n* = 8)_ = 0.33, *p* = 0.76; Winter: r = 0.36, t_(*n* = 8)_ = 0.95, *p* = 0.38; treatments combined: r = 0.20, t_(*n* = 16)_ = 0.77, *p* = 0.46). For individual compounds, treatments combined, we found positive significant correlations between the SCP and glycerol (r = 0.63, t_(*n* = 16)_ = 3.03, *p* < 0.01) and saccharose (r = 0.53, t_(*n* = 16)_ = 2.33, *p* < 0.05). The positive SCP × glycerol correlation also applied in Autumn (r = 0.82, t_(*n* = 8)_ = 3.55, *p* < 0.05). No correlations were found in Winter. Adding the *Erebia* results again returned no SCP × TSPC correlation (Autumn r = 0.02, t_(*n* = 13)_ = 0.08, *p* = 0.94; Winter: r = 0.13, t_(*n* = 12)_ = 0.40, *p* = 0.70; treatments combined: r = 0.06, t_(*n* = 25)_ = 0.31, *p* = 0.76) and no correlation with glycerol, but a significant positive correlation with myo-inositol (treatments combined: r = 0.40, t_(*n* = 25)_ = 2.12, *p* < 0.05) and saccharose (treatments combined: r = 0.44, t_(*n* = 25)_ = 2.44, *p* < 0.05; Autumn: r = 0.56, t_(*n* = 12)_ = 2.27, *p* < 0.05) (Appendix A provides detailed correlation results). 

Across all species for which SCP or TSPC was available (cf. Table 1), SCP did not correlate with elevation of origin (r = 0.184, t_(*n* = 17)_ = −0.72, *p* = 0.48), whereas TSPC and elevation correlated strongly positively (r = 0.854, t_(*n* = 17)_ = 5.45, *p* < 0.001). The same result, with a stronger TSPC × elevation correlation in Winter, held for treatments analyzed separately (Figure 2), implying larger winter buildup of putatively cryoprotective compounds in high-elevation species. 

### 3.2. Sugar and Polyol Profiles

The eight non-*Erebia* species differed in sugar and polyol profiles (Table 2, Appendix A). In Autumn (Figure 3A), the alpine *C. gardetta* and steppe *C. briseis* contained high sorbitol, ribose, trehalose (mainly the former), and glucose (the latter). They contrasted from *M. galathea* with high maltose, fructose, glycerol, and saccharose. The remaining species were intermediate, with the steppe *M. dryas* containing high glucose, and the mesic grassland species *C. arcania*, *A. hyperantus*, and *M. jurtina* containing high ribitol, mannitol, and saccharose. After adding the earlier *Erebia* results (Figure 3B), all *Erebia* spp. formed a loose group, associated with high trehalose (highest in the alpine *E. cassioides*), ribitol, threitol, and erythritol (the latter two unique for *Erebia* spp.). *C. gardetta* and *C. briseis* assumed intermediate positions, displaying association with sorbitol and glucose. The dry grassland species *M. dryas* and *H. semele* also inclined towards sorbitol and glucose, whereas the remaining non-*Erebia* mesic grassland species displayed high concentrations of saccharose, fructose, and myo-inositol. In Winter, glucose and myo-inositol increased in *H. semele*, glucose increased in *M. galathea* (Figure 3C), and trehalose and glycerol increased in *E. pronoe* and *E. aethiops* (Figure 3D). 

The main patterns were retained for both treatments analyzed together, and the treatment effect was considered covariate (Table 2, Appendix A). The mesic grassland/lowland species were still associated with high glucose, maltose, ribose, sorbitol, and fructose, whereas the alpine species contained high trehalose, threitol, erythritol, and ribitol. The secondary distinction between steppe (high glucose, ribose, and sorbitol) and mesic grassland (high saccharose, mannitol, and fructose) species was also retained. Treating species identities as covariates showed that maltose, saccharose, and ribitol were higher in Autumn, whereas arabinitol, erythritol, sorbitol, ribitol, and trehalose were higher in Winter. 

### 3.3. Phylogenetic Signal and Ancestral State Reconstruction

The pruned tree topology corresponded to previous results on Satyrinae butterflies [12,43]. From a common ancestor, the two *Coenonympha* spp. branched off first, and the second division was between (*Erebia* + (*A. hyperantus* + *M. jurtina*)) and *M. galathea* + (*H. semele* + (*M. dryas* + *C. briseis*))). Blomberg’s K values were significant, implying a phylogenetic dependency for Winter glycerol and Winter SCP, and marginally significant for winter TSPC (Table 3). Pagel’s λ was significant only for winter glycerol, but the value was also close to 1 for Winter SCP. 

The ancestral state reconstructions (Figure 4) revealed that Winter glycerol was high in the common ancestor of the assayed species, and decreased in *Coenonympha* spp. and in the steppe representatives *H. semele*, *M. dryas*, and *C. briseis*. SCP was ≈−16 °C in the common ancestor. It decreased in the *Coenonympha* branch, increased in *M. galathea*, *H. semele*, (*A. hyperantus* + *M. jurtina*), and then independently decreased in alpine *Erebia* spp. and (*M. dryas* + *C. briseis*). The total concentration of sugars and polyols was ≈5 μg × mg^−1^ in the common ancestor, decreased in the *Coenonympha* branch, dramatically increased in the branch leading to *Erebia* spp., and again decreased in (*A. hyperantus* + *M. jurtina*). 

### 3.4. Relations of Cold Hardiness Traits to Sugars and Polyol Profiles

Three life-history traits were related to sugars and polyol profiles in the fourth-corner RDA analysis: TSPC (explained variation, F, *p*: 23.7%, 3.4, 0.001), elevation (simple/conditional effects: 18.3%, 2.5, 0.024/13.7%, 2.2, 0.097) and larval instar (15.2%, 2.0, 0.088/10.5%, 1.8, 0.19). In the final model (Figure 5), high-elevation species with high TSPC overwintering in early larval instars were correlated with positive values of CCA1 axis, i.e., with high representation of trehalose, threitol, ribitol, arabinitol, erythritol, and glycerol. Low-elevation and low-TSPC species overwintering in late instars contained high concentrations of fructose, saccharose, myo-inositol, and glucose. 

## 4. Discussion

Expanding the interest of overwintering butterfly larvae cold hardiness from the cold-adapted genus *Erebia* to a broader sample of European univoltine Satyrinae reveals that the high diversity of strategies and mechanisms detected earlier [14,15] is not restricted to the single genus. Among the eight non-*Erebia* species, two survived freezing of their body fluids, four were killed by the freezing, and three displayed a mixed strategy. The supercooling point values varied among all species by >10 °C, and concentrations of sugars and polyols differed fourfold (sixfold in Autumn) and eightfold when considering the previously assayed *Erebia* spp. There was neither a straightforward relationship between per-species SCP and TSPC, nor a clear correlation between concentrations of individual compounds and SCP. Instead, we detected phylogenetic signals in Winter glycerol, Winter SCP, and Winter TSPC. High-elevation species with high TSPC tend to be freeze-avoidant and overwinter in earlier larval instars. During acclimation, they accumulate trehalose and glycerol, as well as threitol and arabinitol, the latter two biosynthesized from glucose via the pentose phosphate pathway [44]. Low-elevation species, on the other hand, tend to overwinter in later instars and with elevated levels of glucose, saccharose, fructose, and maltose, i.e., compounds directly involved in primary metabolism. 

The difference between high- and low-altitude species is unlikely due to larval food, as all the species develop on closely related plants (which can be interchangeable in captive rearing [10]). The overwintering in early larval instars in high-altitude species (and the steppe species *C. briseis*) is probably due time constraint for pre-diapause development, due to short season in the mountains [7] and late-season reproduction in *C. briseis* [28]. 

A sample of eight species (sixteen with the earlier *Erebia* results) assayed may seem high, given that studies of insect cold hardiness rarely sample multiple representatives of well-defined clades (but see [45,46,47,48,49,50]). It is admittedly poor compared to the radiation of temperate Satyrinae [12,51,52], but still allows cursory inference regarding ecological and geographical correlates. 

Comparisons of insect antifreeze strategies across distant [53] and related [45,54] species indicate that ability to survive the freezing of body prevails in regions with oceanic climates (subantarctic islands: [45]; coastal mountains: [20]), which experience short, unpredictable, but frequent frosts. Continental conditions with temporarily predictable and undisrupted freezing periods favor freeze avoidance. In our sample, the connection between oceanicity and freeze tolerance applies for *H. semele* [32], whose coastal populations are prospering at present, whereas inland populations are declining [33]; however, this is also true for *A. hyperantus*, a drought-sensitive species [55,56]. *Erebia medusa*, also displaying this strategy, is peculiar among European congenerics by a broad distribution spanning from lowlands to mountains and from damp to xeric habitats [57]. A mixed strategy, previously reported for *E. aethiops* [15], which shares wide vertical distribution with *E. medusa* [58], was detected here in three additional species. *Minois dryas* inhabits a wide Euro-Siberian range with various climatic conditions; moreover, in Central Europe it inhabits both dry and humid habitats [34]. *Chazara briseis* is a specialist of continental steppes [28,29], whereas *M. jurtina* is the most common European grassland butterfly, persisting even in intensively exploited landscapes [59,60]. Other lepidopteran examples displaying mixed responses to freezing of body fluids are *Papilio zelicaon* (Lucas, 1858) (Lep.: Papilionidae), a North American species distributed from the humid coast to arid inland [16], and *Pieris rapae* (Linnaeus, 1758) (Lep.: Pieridae), a multivoltine generalist distributed from subtropical to arctic regions [61]. The mixed strategy thus appears beneficial in variable conditions, climatic or otherwise, among seasons or even within an individual lifespan. Climate variation among seasons is certainly the case of continental steppes, with prominent inter-seasonal variation in snow cover and temperature [62]; recall that *C. briseis* larvae may feed during mild winters. From all species assayed, *C. briseis* and *M. jurtina* also displayed the largest variation in SCP between seasonal treatments, indicating flexible reactions to autumn conditions. *E. aethiops* displays local adaptations in adult thermoregulation [63]; other adaptations, variable among individuals, might exist in larval cold hardiness. 

All the Satyrinae species investigated here displayed lower concentrations of sugars and polyols than the alpine *Erebia* representatives. Still, the highest-elevation *E. pluto* with concentrations ≈18 μg × mg^−1^ is an exception among Lepidoptera. The overwintering pupae of the cabbage white, *Pieris brassicae* (Linnaeus, 1758), may contain ≈20 μg × mg^−1^ of its major cryoprotectant, sorbitol [64], and those of the green-veined white, *Pieris napi* (Linnaeus, 1758), ≈30 μg × mg^−1^ of total sugars [65]. Overwintering adults of *Inachis io* (Linnaeus, 1758) (Lep.: Nymphalidae) and *Aglais urticae* (Linnaeus, 1758) (Lep.: Nymphalidae) both reach ≈30 μg × mg^−1^ of the main cryoprotectant, glycerol [66]. In overwintering *Colias palaeno* (Linnaeus, 1761) (Lep.: Pieridae) larvae, a still higher concentration (≈60 μg × mg^−1^) of total sugars and polyols was found in a mountain-zone but not an alpine-zone population [67]. The pyralid moth, *Apomyelois ceratoniae* (Zeller, 1839), a subtropical orchard pest, reaches ≈120 μg × mg^−1^ for glycerol, sorbitol, and glucose combined [68]. A record value for Lepidoptera was found in the tortricid moth, *Choristoneura fumiferana* (Clemens, 1865), reaching ≈150 μg × mg^−1^ glycerol [69]. High sugar and polyol concentrations, however, are not necessary for dwelling in (sub)alpine habitats, as alpine *C. gardetta* had similarly low TSPC as its lowland congener, *C. arcania*. 

It should be emphasized that the sugars and polyols assayed here did not surprisingly display, either in total concentrations or separately, a straightforward cryoprotective function, in terms of decreasing supercooling point. The same holds for the *Erebia* representatives analyzed separately, in which glycerol was always low, whereas saccharose was high despite cold hardiness being low (=SCP high; [15]). These carbohydrates can play other protective roles, such as stabilization of macromolecules, at much lower concentrations than are needed for changes in SCP [70]. In contrast, a study of two *Colias* butterflies detected high glycerol and a tight negative correlation between glycerol and SCP, i.e., a straightforward positive effect of glycerol on cold hardiness [71]. In addition to this example, glycerol functions as a major cryoprotectant in a wide range of insects (e.g., *Xylotrechus rusticus* (Linnaeus, 1758) (Coleoptera: Cerambycidae) [72]; *Eurygaster integriceps* (Puton, 1881) (Hemiptera: Scutelleridae) [73]), including Lepidoptera (e.g., *Chilo suppressalis* (Walker, 1863) (Lep.: Pyralidae) [74], *Choristoneura fumiferana* (Clemens, 1865) [69], *Parnassius bremeri* Bremer, 1864 (Lep.: Papilionidae) [75]). Its role may vary within species, as shown on *C. palaeno* larvae from two elevations [67]. Another compound correlated with SCP, but at low concentration, was myo-inositol, a major cryoprotectant in the dermestid beetle, *Trogoderma granarium* (Everts, 1898) [76], and ladybird, *Ceratomegilla undecimnotata* (Schneider, 1792) [77].

The conflicting information implies that individual compounds fulfill different roles among taxa and during phases of overwintering [78]. What matters eco-evolutionarily is the functional outcome, expressed here as the value of SCP. Our results suggest that the SCP reflects a phylogenetic signal (or convergent evolution) towards low values in alpine *Erebia* spp., and in species of continental steppe environments. A special case was the two *Coenonympha* representatives, the subalpine *C. gardetta* and lowland *C. arcania*, both displaying low SCP but also low concentrations of putative cryoprotectants. Possibly, larvae of this genus, phylogenetically most distant to the remaining species [31], employ other cryoprotective agents, such as amino acids and antifreeze proteins. The case of the subtropical orchard pest *A. ceratoniae* indicates that a species experiencing rapid and unpredictable changes of winter weather can be supremely protected against freezing [68]. The low SCP in *C. arcania* could be a pre-adaptation of its ancestor for colonization of high elevations by its descendants (cf. [31]). 

The evolutionary signals in glycerol concentration, TSPC, and SCP were apparent only in the Winter treatment. The conditions experienced by pre-hibernation larvae likely vary with species and habitats, and diverse external signals may launch cryoprotection [71]. Phasing of diapause is crucial for cold hardiness development [79]. Constitutive (Autumn) cryoprotectants can be replaced by inducible (Winter) ones [77]. Although we found some Autumn vs. Winter changes in the cryoprotectant profiles, there was no clear pattern allowing to distinguish between constitutive and inducible compounds, or between the role of polyols vs. sugars.

High trehalose and several polyols occurred in alpine species (*Erebia* spp., *C. gardetta*), in which they correlated with elevation, but also in the steppe *C. briseis*. Across biotic realms, trehalose enhances cryoprotection in extreme environments [80,81]. However, its high content did not decrease SCP among our species. It more likely protects cells against desiccation [82], which may be critical both at dry steppes (*C. briseis*) and wind-exposed alpine cliffs (*E. pluto*, *E. cassioides*) (cf. [83]). On the other hand, mono- and disaccharides were more typical for low-elevation species. Among them, inhabitants of dry grasslands (*M. dryas*, *C. briseis*, *H. semele*) contained high concentrations of sorbitol, glucose, and ribose, whereas mesic habitat species (*M. galathea*, *M. jurtina*, *C. arcania*, *A. hyperantus*) contained high saccharose, maltose, and mannitol.

The biochemical pathways synthesizing and assimilating individual polyols are closely related [84]. Individuals modify both cold hardiness and polyol profiles during the season, and variation among populations exists [67]. Given the diversity of cold hardiness strategies among temperate Satyrinae, contrasting with their uniform feeding on grasses and developmental modes (overwintering as larvae), it is tempting to speculate whether adaptation to varying climatic niches could have propelled the evolution of their remarkable species and habitat use diversity.

## Figures and Tables

**Figure 1 insects-13-00369-f001:**
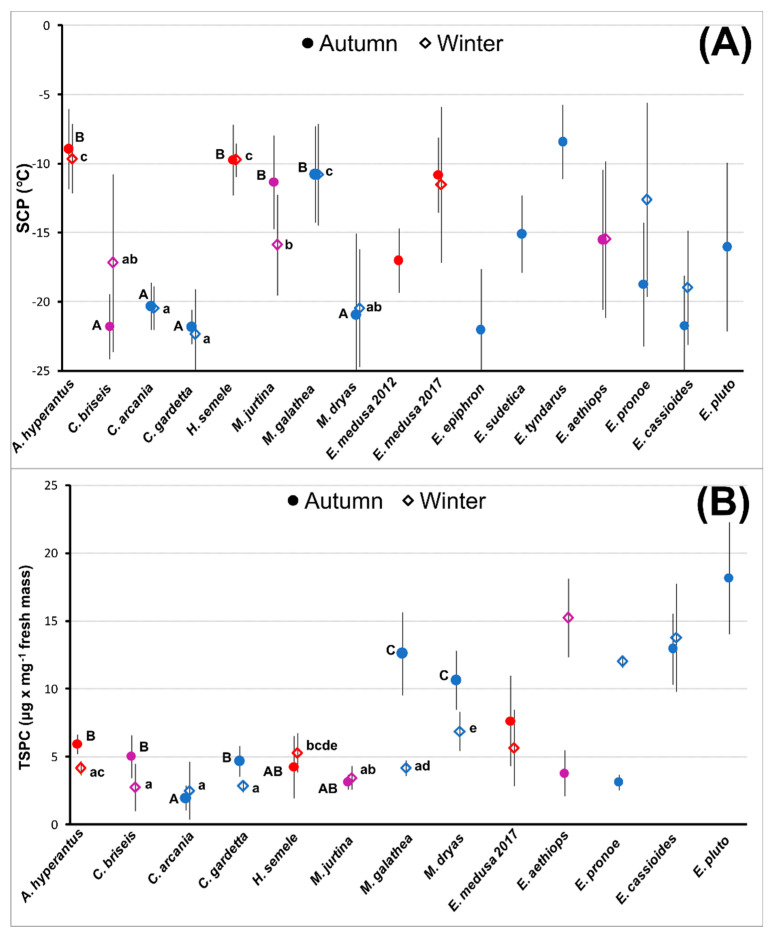
Overview of supercooling points, SCP (**A**), and total sugars and polyols concentrations, TSPC (**B**), found for the eight Satyrinae species assayed for this study, and the eight *Erebia* spp. assayed in [14,15]. Filled circles stand for Autumn, and empty diamonds for Winter treatments. The letters accompanying the marks denote significant differences among species and treatments, as revealed for the eight newly assayed Satyrinae species by Tukey’s HSD test (Appendix A). Color codes: freeze-avoidant species are blue, freeze-tolerant species red, mixed-strategy species violet.

**Figure 2 insects-13-00369-f002:**
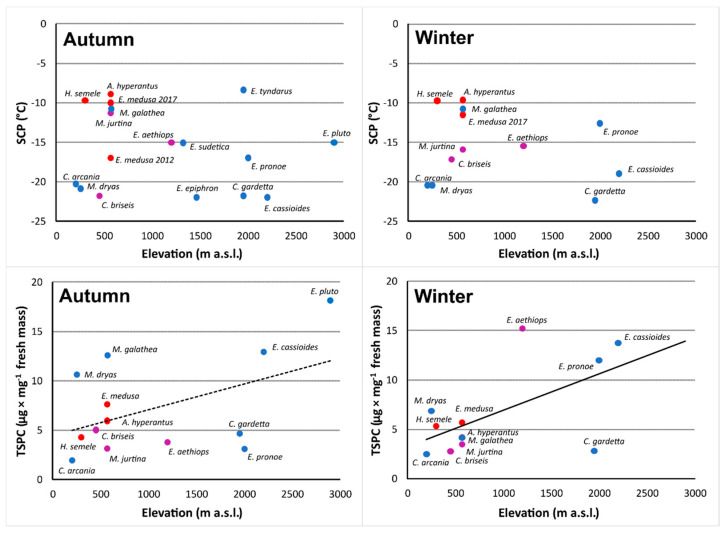
Relating average supercooling points (SCP) and total sugar and polyol concentrations, TSPC, detected for European Satyrinae species, to elevations of their origin. SCP did not correlate with elevation (Autumn: r = −0.163, t_(*n* = 17)_ = −0.64, *p* = 0.53; Winter: r = −0.25, t_(*n* = 12)_ = −0.82, *p* = 0.43). A dashed line shows a tendency towards positive correlation (r = 0.48, t_(*n* = 13)_ = 1.80, *p* < 0.1), a solid line shows a significant positive correlation (r = 0.60, t_(*n* = 12)_ = 2.37, *p* < 0.05). Color codes: freeze-avoidant species (unable to survive freezing of body fluids) are blue, freeze-tolerant species (surviving freezing of body fluids) red, mixed-strategy species (partly surviving, partly not) violet.

**Figure 3 insects-13-00369-f003:**
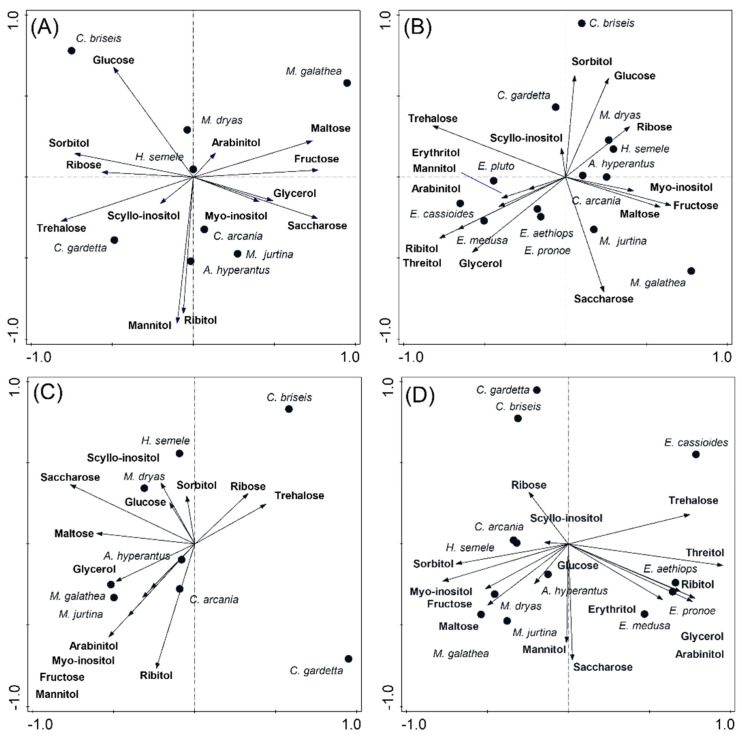
CCA ordination biplots, showing the distributions of putatively cryoprotectant compounds in overwintering Satyrinae butterfly larvae. (**A**,**C**): the eight species considered in this study. (**B**,**D**): the analyses expanded by adding the five *Erebia* species from [15]. (**A**,**B**): Autumn treatment analyzed separately. (**C**,**D**): Winter treatment analyzed separately. See Appendix A for the treatments analyzed together.

**Figure 4 insects-13-00369-f004:**
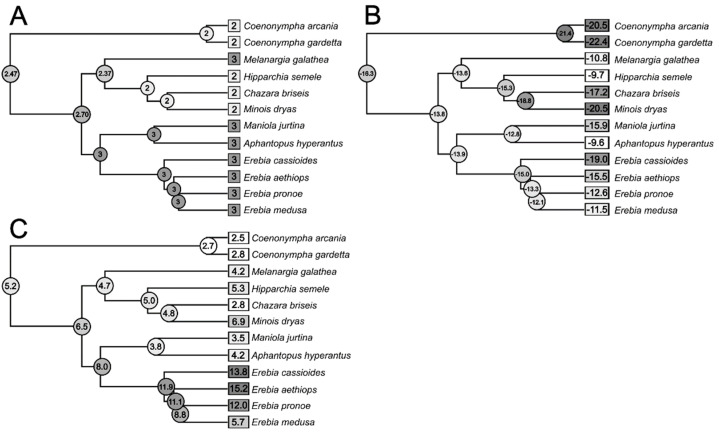
Phylogenetic trees with mapped ancestral states of (**A**) Winter glycerol (log μg/g), (**B**) average supercooling points (SCP, °C), and (**C**) Winter total sugar and polyol concentrations (TSPC, μg × mg^−1^) in larvae of selected European Satyrinae. Measured values are depicted on branch tips, and ancestral states, inferred by the phylogenetically independent contrasts, are shown on each node of the tree. For tree inference, see Appendix B.

**Figure 5 insects-13-00369-f005:**
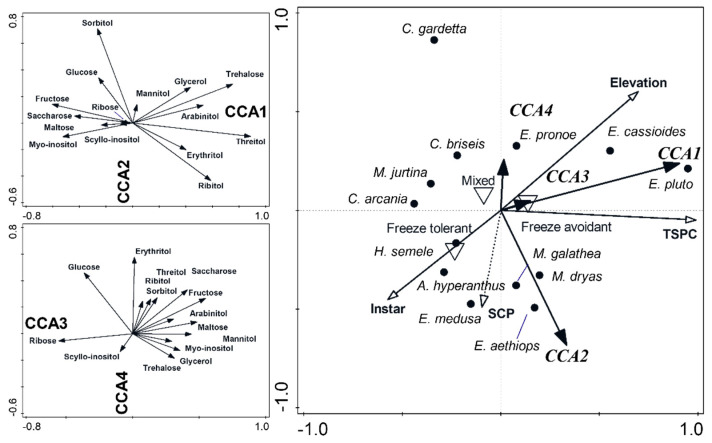
Relating cold hardiness and overwintering-related traits to sugars and polyol profiles of overwintering larvae of European monovoltine Satyrinae. The two left panels show CCA ordination species ≈ compounds|treatment (eigenvalues 0.906, 0.832, 0.792, 0.692; adjusted explained variation 40.6%, F/P_1st axis_ = 18.9/0.001, F/P_all axes_ = 12.2/0.001). The panel in the right is a triplot, showing the relation of the CCA axes (thick black darts) from the left panel to the traits. Traits significantly related to the axes are shown as narrow full darts; those without significant contribution (depicted as supplementary variables) as dotted darts or white triangles. The right panel model eigenvalues: 0.239, 0.148, 0.092, 0.250; explained variation 30.5%; F/P_1st axis_ = 2.8/<0.01, F/P_all axes_ = 2.8/<0.01.

**Table 1 insects-13-00369-t001:** Overview of the Satyrinae butterflies analyzed, including *Erebia* butterflies from earlier studies [14,15]. Localities and dates of sampling, their respective elevation, and numbers (*n*) of larvae used to assess supercooling point (SCP), lower lethal temperature (LLt), and total sugar and polyol concentration (TSPC). * The two numbers separated by “/” denote the numbers of larvae used for Autumn/Winter treatment.

*Species* (Phenomena Studied)	Origin (CZ—Czechia, AT—Austria)	Elevation	*n* (SCP) *	*n* (LLt)	*n* (TSCP) *
**This study (SCP, TSPC)**					
*Apanthopus hyperantus*	CZ, Český Krumlov, 48°50′ N, 14°19′ E, July 2018	570 m	16/16	–	10/10
*Chazara briseis*	CZ, Raná, 50°24′ N, 13°46′ E, August 2014–18	450 m	16/14	–	10/10
*Coeneonympha arcania*	CZ, Hodonínská Dúbrava, 48°53′ N, 17°6′ E, June 2018	200 m	16/16	–	10/10
*Coeneonympha gardetta*	AT, Heiligenblutt, 47°3′ N, 12°47′ E, August 2018	1950 m	15/10	–	10/10
*Hipparchia semele*	CZ, Prokopské Údolí, 50°2′ N, 14°21′ E, August 2018	300 m	15/10	–	10/10
*Maniola jurtina*	CZ, Český Krumlov, 48°50′ N, 14°19′ E, July 2018	570 m	16/16	–	10/10
*Melanargia galathea*	CZ, Český Krumlov, 48°50′ N, 14°19′ E, July 2018	570 m	16/16	–	10/10
*Minois dryas*	CZ, Lázně Bohdaneč, 50°4′ N, 15°41′ E, August 2018	250 m	15/8	–	10/8
**Ref. [15] (SCP, TSPC)**					
*Erebia medusa*	CZ, Český Krumlov, 48°50′ N, 14°19′ E, May 2015	570 m	16/16	–	10/10
*Erebia aethiops*	AT, Tirol, Au, 47°06′ N, 10°57′ E, August 2015	1200 m	16/16	–	10/10
*Erebia pronoe*	AT, Pfafflar, 47°17′ N, 10°39′ E, August 2015	1200 m	16/6	–	10/10
*Erebia cassioides*	AT, Hochgurgl, 46°54′ N, 11°03′ E, August 2015	2200 m	16/12	–	10/10
*Erebia pluto*	AT, Rettenbachgletscher, 46°56′ N, 10°55′ E, August 2015	2900 m	15/–	–	10/–
**Ref. [14] (SCP, LLt)**					
*Erebia medusa*	CZ, Český Krumlov, 48°50′ N, 14°19′ E, May 2010	570 m	16	10	–
*Erebia sudetica*	CZ, Praděd Mt., 50°4′ N, 17°13′ E, August 2010	1320 m	16	10	–
*Erebia epiphron*	CZ, Praděd Mt.,50°4′ N, 17°13′ E, August 2010	1460 m	16	10	–
*Erebia tyndarus*	AT, Sölden, Windachtal, 46°57′ N, 11°3′ E, August 2010	1950 m	16	10	–

**Table 2 insects-13-00369-t002:** Results of canonical correspondence analyses testing the effects of species and treatment (Autumn vs. Winter) on the composition of sugar and polyol profiles of the Satyrinae species assayed in this study, and results obtained after merging the data with five *Erebia* species studied from [15]. Eig1–Eig4 are values of canonical eigenvalues, Var. is variation in the data explained by the ordination model, while F and *p* values refer to results of Monte Carlo permutation tests.

Eight Non-*Erebia* Satyrinae	Eig1	Eig2	Eig3	Eig4	Var.	F, P_1st axis_	F, P_all axes_
Species	0.292	0.154	0.100	0.022	58.2%	62.0 ***	32.2 ***
Treatment	0.051				4.5%	8.4 **	
Species|treatment	0.293	0.153	0.100	0.022	61.8%	66.6 ***	36.8 ***
Treatment|species	0.052				12.5%	22.4 ***	
**Added 5 *Erebia* spp**. [15]							
Species	0.326	0.124	0.065	0.047	63.1%	9.5 ***	33.5 ***
Treatment	0.034				3.4%	8.6 ***	
Species|treatment	0.328	0.123	0.068	0.049	65.9%	10.0 ***	37.7 ***
Treatment|species	0.039				10.7%	27.9 ***	

**: *p* < 0.01; ***: *p* < 0.001.

**Table 3 insects-13-00369-t003:** Results of Blomberg’s K and Pagel’s λ testing the phylogenetic signal in major cryoprotectants, supercooling points (SCP) and total sugar and polyol concentrations (TSPC). See Appendix B for description of inference of the phylogenetic tree, and Figure 4 for mapping the traits onto the tree.

	Autumn				Winter			
Trait	K	*p*	λ	*p*	K	*p*	λ	*p*
Glycerol	0.47	0.40	0.22	0.77	1.97	<0.001	1.11	<0.0001
Fructose	0.38	0.74	<0.01	1.00	0.47	0.50	<0.01	1.00
Glucose	0.45	0.51	0.31	0.74	0.32	0.79	<0.01	1.00
Sucrose	0.51	0.36	<0.01	1.00	0.23	0.96	<0.01	1.00
Trehalose	0.36	0.77	<0.01	1.00	0.42	0.61	<0.01	1.00
SCP	0.69	0.11	<0.01	1.00	0.94	0.03	0.79	0.15
TSPC	0.45	0.50	<0.01	1.00	0.77	0.07	0.42	0.53

## Data Availability

The data on SPC and survival of overwintering larvae are provided in Appendix A, and the mean (±SD) concentrations of sugars and polyols in Appendix A. More detailed data can be provided on request by the corresponding author.

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
