# Peer review of "Exploring Cold Hardiness within a Butterfly Clade: Supercooling Ability and Polyol Profiles in European Satyrinae"

_insects, 2022, doi:10.3390/insects13040369_

Round 1
Reviewer 1 Report
Dear authors, please find my comments and suggestions attached
Best regards

Author Response
The paper is original and is of importance to readers. The authors tried to understand the physiological adaptations and the supercooling capacities of eight Nymphalidae species in comparison with a pilot sample, Erebia. They found phylogenetic and also geographically based differences in the amount of low-molecular-weight carbohydrates and supercooling points of the
species. Overall, with interest, I have read this manuscript which is overall well-written, well-structured and clear. the presentation of the methods and characterization of the results achieved indicate that the method is quite suitable.
R - Thank you for your positive evaluation.
However, I have some more concerns as below:
1- Line 12: Please change to “cold-adapted species and species-rich group of the sub-family Satyrinae”
R - Done.
1- Please add a brief methodology
R - due to limited space, we only added "via gas chromatography – mass spectrometry" to line 25
2- Line 22: Please change to “The cold hardiness of overwintering stages affects the distribution of temperate”
R - Done.
3- Line 25: supercooling points (SCPs)
R - Done.
4- Line 34: what do you mean by “trehalose-based”
R - We changed this to clearer "and contain high concentrations of trehalose" (lines 33-34)
1- Line 41: Please change to “The recent debates on the effects of a warming climate on biotic communities raise the interest.
R - Done.
2- Line 45: the adult season
R - Done.
3- Line 46: in the cold season
R - Done.
4- Line 67: with overwintering is found in Erebia
R - Done.
5- Line 74: a wide range of temperate zone habitats
R - Done.
6- Line 79: on the duration
R - Done.
7- Line 81: rather a small amount
R - Done.
8- Line 87: Knowledge of their specific roles
R - Done.
9- Line 105: play a cryoprotective role
R - Done.
10- Line 107: for the harshness
R - Done.
Line 114: distributed from the Pyrenean peninsula
R - Corrected.
1- Line 119: North of the Alps, its habitat is open-turf dry
R - Done.
2- Line: 158: the Czech Republic
R - Done.
3- Line 209: the elevation of origin
R - Done.
4- Line 210: The composition of sugars
R - Done.
5- Line 28: We used the forward-selection procedure
R - Done.
Overall, the results are well presented but in this section, please consider that all the scientific names of the studied species must be italic with abbreviated to just their initial genus name.
R - Here, we thank for your patience. The italics are embarasing; this somehow happened during uploading the manuscript into the Journal`s template, which is not an apology. We did our best to correct all this mistakes.
Line 252: In both autumn and winter treatments
R - Corrected.
Line 252: A. hyperantus and H. semele larvae survived the body
R - Corrected.
Line 253: No C. arcania, C. gardetta, or M. galathea, larva
R - Corrected.
Line 260: (Figure 1A, Supplementary Table S1)
Line 287 and 291: (Supplementary Table S1)
R - Corrected.
Line 338: and the treatment effect was considered a covariate
R - Done.
Line 388: Three life-history traits
R - Corrected.
Line 394: contained high concentrations of fructose, saccharose, myo-inositol, and glucose
R - Corrected.
Discussion.
You truly mentioned a substantial difference between cryoprotectants of high and low elevation species. I would like to know if this difference may be related to their food?
Overwintering. I would like to know the possible reasons (e.g., ambient temperature, quality, the abundance of food, photoperiod…). Please discuss if possible
R - These are interesting questions, which, we have to admit, did not come to our minds while writing the originally submitted versions. The answers are rather obvious, we tackle them both at lines 496-500, and they are linked to basic life history-habitats relations (the alpine species do not have time to produce bigger overwintering larvae, and the food is, chemically, very similar across all Satyrinae). The discussion on the topics could be more elaborate, but it would disrupt the current flow of the text. We hope that our solution is satisfactory.
Line 433: favor freeze avoidance
R- corrected
Avoid referencing to Tables and Figures in this section
R - We deleted the few refs to Tables and Figures.
5. Line 439 and elsewhere some scientific names remained unitalic
R - see the related answer above; corrected.
6. Line 440 and elsewhere please write the full scientific names of the species for the first time.
R - Done (we consider full name as having 4 parts, incl. the year of description)
7. Line Line 445 and elsewhere please write the scientific name of the species for the first time like this: Papilio zelicaon (Lucas)(Lep.: Papilionidae)
R - done, see also the above response.
8. Line 454: E. aethiops
R - corrected
9. Line 461: Overwinter- 461ing adults of Inachis io (L.) and Aglais urticae (L.) both reach ≈30 μg × mg-1 of ???? Please specify
R - added "of the main cryoprotectant, glycerol" (now at line 545)
10. Line 462: still higher concentration of ?????
R - "of total sugars and polyols" (now at line 457; we apologize for these two omissions)
11. Line 465: The pyralid moth,
12. Line 467: in the tortricid moth,
R - both above mistakes corrected
13. Line 468: High sugars and polyols concentrations
R - changed as suggested
14. Line 474: was analysed
R - changed as suggested
Line 481: Coleoptera
R - typo corrected
Line 482: Please remove “including”
R - done
Line 485: C. palaeno
R - changed as suggested.
Line 486: but at low concentration
Line 487: in the dermestid beetle,
Line 487: and ladybird,
R - all the above mistakes corrected.
22. Line 493: the special case was the two Coenonympha representatives, the subalpine C. gardetta and lowland C. arcania, both displaying low SCP, but also low concentrations of putative cryoprotectants. What are the possible reasons? Maybe other cryoprotective agents such as amino acids and antifreeze proteins are accumulated????
R - we basically accepted your hint, and added reference to phylogeny, which shows that Coenonympha are quite distant evolutionarily from all the remaining species studied.
23. Line 496: the subtropical orchard pest, A. ceratoniae
R - corrected as suggested.
Line 520: the season,
R - corrected as suggested.
R - corrected as suggested.
Line 520: among populations exist
Reviewer 2 Report
I strongly recommend the paper to be published
Author Response
We thank you for the positive review.
Reviewer 3 Report
The manuscript, “Exploring cold hardiness within a butterfly clade: supercooling ability and polyol profiles in European Satyrinae”, by Vrba et al. explores the role of cold hardiness among Satyridae butterflies. Although their findings do not suggest a straightforward cryoprotective function, they do suggest, however, that super cooling point reflects a phylogenetic signal. The combination of previous dataset on Erebia butterflies and their current pool of dataset increase the robustness of their results, help the authors to better explain the ecological patterns observed and crystalize their findings under an evolutionary point of view. In the light of further environmental change and the urgent need to understand how species will evolve and respond to shifts in temperature, snow cover and extreme weather events such as warm winters, this study is of great interest and importance for the scientific community.
Overall, the manuscript is well-written, literature’s up-to-date and authors’ experience and knowledge of the target system is noticeable. I would like to point out how cleanly written the manuscript is, especially the methodological part, which will be of great help to the readers, if decide to follow these or similar statistical approaches. The availability of SCP and TSCP species traits will serve as the foundation of the studies dealing with pre-hibernation larvae on cold hardiness. I have a couple of points the authors may want to consider.
Lines 179-187: Although the description here satisfies the purpose of the study, any additional optical material (video, photos) of the process it would help the readers to better visualize the whole process.
Lines 189-190: Please add a short description of sample preparation also here.
Lines 258-259: Is it also possible to provide the survival for Erebia as well?
Lines 318-319: You might want to explain what freeze-avoidant and freeze-tolerant species are here.
Line 528: change to “measurements”.
-Sometimes it gets hard to connect supercooling ability and the role of contents of sugars and polyols. A nice example is given in Lines 418-422. It might be helpful to provide a table in Appendix or SI where you’d give this kind of linkage support between contents of sugars/polyols and ecological/evolutionary aspects of the studied organisms.
-I believe that phylogenetic signal is a clear and novel pattern you get. You might want to consider including it in your title.
Author Response
Dear reviewer.
We thank you for your positive and thoughtful review. Let us now response to your comments one-by-one.
Lines 179-187: Although the description here satisfies the purpose of the study, any additional optical material (video, photos) of the process it would help the readers to better visualize the whole process.
R - It would absolutely help, but being a bit conservative (I still like most the "boringly factual" black&white ink drawings of papers of my early career), I opt not to expand the manuscript by additional material. We promise, however, that we will think about this options for the next time.
Lines 189-190: Please add a short description of sample preparation also here.
R - The original mentioning of "procedure" was redundant and a bit silly here, as it referred to [37], in which nothing more is written. Petr Simek, one of the co-author (and the most chemically skilled of all of us) personally apologizes for this. We corrected the sentence, see line 197.
Lines 258-259: Is it also possible to provide the survival for Erebia as well?
R - Of course, we added it, see line 268.
Lines 318-319: You might want to explain what freeze-avoidant and freeze-tolerant species are here.
R - Done.
Line 528: change to “measurements”.
R - Done.
-Sometimes it gets hard to connect supercooling ability and the role of contents of sugars and polyols. A nice example is given in Lines 418-422. It might be helpful to provide a table in Appendix or SI where you’d give this kind of linkage support between contents of sugars/polyols and ecological/evolutionary aspects of the studied organisms.
R - It is a good suggestion, but it would represent a research task on its own. We are considering it, for a future.
-I believe that phylogenetic signal is a clear and novel pattern you get. You might want to consider including it in your title.
R - We thank you for this compliment, but opt not to do so, for two reasons. First, the taxon sampling is too incomplete for such a bold conclusion, and second, the phylogeny is in the title, implicitly, in the term "Clade". It should suffice for us, cladistically oriented "inner circle".